# Interpretable machine learning-based individual analysis of acute kidney injury in immune checkpoint inhibitor therapy

**Minoru Sakuragi**[1,2], **Eiichiro Uchino**[1,2], **Noriaki Sato**[1,2], **Takeshi Matsubara**[2],
**Akihiko Ueda**[1,3], **Yohei Mineharu**[1,4,5], **Ryosuke Kojima**[1], **Motoko Yanagita**[2,6]*,
**Yasushi Okuno**[1]*

1 Department of Biomedical Data Intelligence, Graduate School of Medicine, Kyoto University, Kyoto, Japan,
2 Department of Nephrology, Graduate School of Medicine, Kyoto University, Kyoto, Japan, 3 Department of
Gynecology and Obstetrics, Graduate School of Medicine, Kyoto University, Kyoto, Japan, 4 Department of
Neurosurgery, Graduate School of Medicine, Kyoto University, Kyoto, Japan, 5 Department of Artificial
Intelligence in Healthcare and Medicine, Graduate School of Medicine, Kyoto University, Kyoto, Japan,
6 Institute for the Advanced Study of Human Biology (ASHBi), Kyoto University, Kyoto, Japan

* okuno.yasushi.4c@kyoto-u.ac.jp (YO); motoy@kuhp.kyoto-u.ac.jp (MY)

org/10.1371/journal.pone.0298673

Farmacologiche Mario Negri, ITALY

**Data Availability Statement:** Data cannot be
shared publicly because of patient privacy in
electronic medical records. Data are available from
Kyoto University Graduate School and Faculty of

## Abstract

### Background

Acute kidney injury (AKI) is a critical complication of immune checkpoint inhibitor therapy.
Since the etiology of AKI in patients undergoing cancer therapy varies, clarifying underlying
causes in individual cases is critical for optimal cancer treatment. Although it is essential to
individually analyze immune checkpoint inhibitor-treated patients for underlying pathologies
for each AKI episode, these analyses have not been realized. Herein, we aimed to individu-
ally clarify the underlying causes of AKI in immune checkpoint inhibitor-treated patients
using a new clustering approach with Shapley Additive exPlanations (SHAP).

### Methods

We developed a gradient-boosting decision tree-based machine learning model continu-
ously predicting AKI within 7 days, using the medical records of 616 immune checkpoint
inhibitor-treated patients. The temporal changes in individual predictive reasoning in AKI
prediction models represented the key features contributing to each AKI prediction and clus-
tered AKI patients based on the features with high predictive contribution quantified in time
series by SHAP. We searched for common clinical backgrounds of AKI patients in each
cluster, compared with annotation by three nephrologists.

### Results

One hundred and twelve patients (18.2%) had at least one AKI episode. They were clus-
tered per the key feature, and their SHAP value patterns, and the nephrologists assessed
the clusters' clinical relevance. Receiver operating characteristic analysis revealed that the
area under the curve was 0.880. Patients with AKI were categorized into four clusters with
significant prognostic differences (p = 0.010). The leading causes of AKI for each cluster,

Medicine, Ethics Committee via email (ethcom@kuhp.kyoto-u.ac.jp) or telephone (+81-75-753-4680) for researchers who meet the criteria for access to confidential data.

**Funding:** The authors received no specific funding for this work.

**Competing interests:** The authors have declared that no competing interests exist.

such as hypovolemia, drug-related, and cancer cachexia, were all clinically interpretable, which conventional approaches cannot obtain.

## Conclusion

Our results suggest that the clustering method of individual predictive reasoning in machine learning models can be applied to infer clinically critical factors for developing each episode of AKI among patients with multiple AKI risk factors, such as immune checkpoint inhibitor-treated patients.

## Introduction

Acute kidney injury (AKI) is a critical complication with significant prognostic implications often observed in cancer patients [1–3]. Immune checkpoint inhibitors (ICIs) are key therapeutic agents for advanced cancer that can cause renal-related adverse events during their administration, including AKI [4–8]. With the increasing use of ICIs, the incidence of AKI during ICI therapy has been reported to be as high as 14–18% [9–11]. The development of AKI during systemic therapy, such as ICI therapy, not only increases the risk of death and the adverse effects on multiple organs but also represents a major cause of interruption of cancer treatment [1, 2]. Several risk factors for AKI, including baseline renal function, proton pump inhibitors (PPI), and immune-related adverse events (IrAEs), have been reported in ICI-treated patients [12–17]. However, these studies analyzed the population as a whole and did not perform individual risk analyses for each AKI episode in each patient. Since the etiology of AKI in patients undergoing cancer therapy varies, even among those diagnosed with the same type of AKI, clarifying the causes of AKI is critical for achieving optimal cancer treatment. Therefore, it is essential to individually analyze ICI-treated patients for existing underlying pathologies causing the onset of each episode of AKI. However, these individual analyses have not yet been realized with conventional clinical research methods, and no such study has been reported.

Herein, we investigated the underlying background of AKI in ICI-treated patients by applying a new approach to classify and analyze time-series individual predictive reasoning of machine learning (ML)-based AKI prediction models. First, we focused on the fact that the temporal changes in individual predictive reasoning in continuous AKI prediction models represent the key features contributing to each AKI prediction. We then estimated that in AKI prediction models, patients with similar predictive reasoning shared similar underlying factors for AKI development, and clustered AKI patients based on the pattern of features with high predictive contribution quantified in time-series by SHapley Additive exPlanations (SHAP) [18]. Thus, we compared each cluster with nephrologist chart review findings, which revealed crucial underlying factors involved in AKI development in individual ICI-treated patients that were not previously observed. Furthermore, the predictive reasoning consisted of combinations of features reasonably interpretable by clinicians.

Our results enabled us to clarify the background of AKI development in ICI-treated patients with underlying risks for AKI and suggested the potential for medical applications of ML prediction models as interpretable artificial intelligence (AI) to medical care, which had been a challenge to explainability.

## Materials and methods

### Model development and definitions

We created a dataset from the electronic medical records (EMRs) of 616 patients who received ICI therapy for cancer at the Kyoto University Hospital from July 2014 to September 2019 and constructed an AKI prediction model. Using this dataset, we constructed an ML-based model to continuously predict the development of AKI within 7 days of the reference date (S1 and S2 Figs in S1 File). Subsequently, we visualized the predictive reasoning among patients with AKI using SHAP and evaluated the clinical validity of patient clustering using predictive reasoning for AKI development (Fig 1). AKI was defined based on serum creatinine (SCr) changes ($\geq$ 0.3 mg/dL or 1.5 times increase from baseline) according to Kidney Disease: Improving Global Outcomes diagnostic criteria [19] (S1 Method in S1 File). The period for the prediction model was defined as the period from the ICI initiation in each patient to the end of December 2019; patients with multiple AKI events within 14 days from the date of the first episode of AKI were excluded from the evaluation.

We used LightGBM [20], a gradient-boosting decision tree, as a prediction algorithm to build a classification model that would continuously predict AKI within 7 days from each time point (Fig 1). The main reasons for selecting LightGBM were its flexibility in handling medical records that potentially contain a certain number of missing values and its ability to perform high-speed calculations (S1 Table in S1 File). We used 287 clinical variables obtained from EMRs as input features for each patient (S2 Method in S1 File). For the features linked to time series, data from the 4 weeks before the reference date were divided into four windows, one for each week, and each window was labeled "(-1 wk)," "(-2 wk)," "(-3 wk)," and "(-4 wk)," and suffixes were assigned to each feature (S1 Fig in S1 File). In addition, the objective variable was labeled "AKI-positive" if the patient developed AKI within 7 days of the predicted time point (S2 Fig in S1 File). All analyses were conducted using Python 3.7.7 (https://www.python.org/doc/), with scikit-learn [21] 0.22.1 (https://scikit-learn.org/stable/index.html#) and LightGBM 2.3.0 (https://lightgbm.readthedocs.io/en/stable/#) libraries for model development, and statsmodels 0.13.2, rpy2 3.5.2, and lifelines 0.25.9 libraries for statistical analysis.

### Visualizing individual AKI predictive reasoning and clustering

SHAP is a game theory-based model interpretation framework that quantitatively evaluates the contribution of each input feature as a SHAP value [18]. Unlike previous studies, we performed a unique visualization in which SHAP values at all prediction time points were arranged in a time series (Fig 1). The SHAP method was implemented using the Python SHAP package (https://shap.readthedocs.io/en/latest/).

We performed hierarchical clustering for patients with AKI based on the patterns of SHAP values and searched for common clinical backgrounds in each cluster. Subsequently, we compared the clinical backgrounds in each cluster with AKI causes, as annotated by three nephrologists (S3 Fig in S1 File). All chart reviews and the free-text annotations of the nephrologists for AKI causes were conducted independent of ML model analysis and without being influenced by each other (S3 Method in S1 File). Furthermore, we evaluated the clinical validity of the clustering by observing the 90-day survival after the first episode of AKI with the Kaplan—Meier analysis. In addition, categorical variables and means among clusters were compared using Fisher's exact probability and Kruskal—Wallis tests, respectively. Finally, the distribution of annotation labels within each cluster was evaluated using Chi-square goodness-of-fit test. Statistical significance was defined as $p < 0.05$.

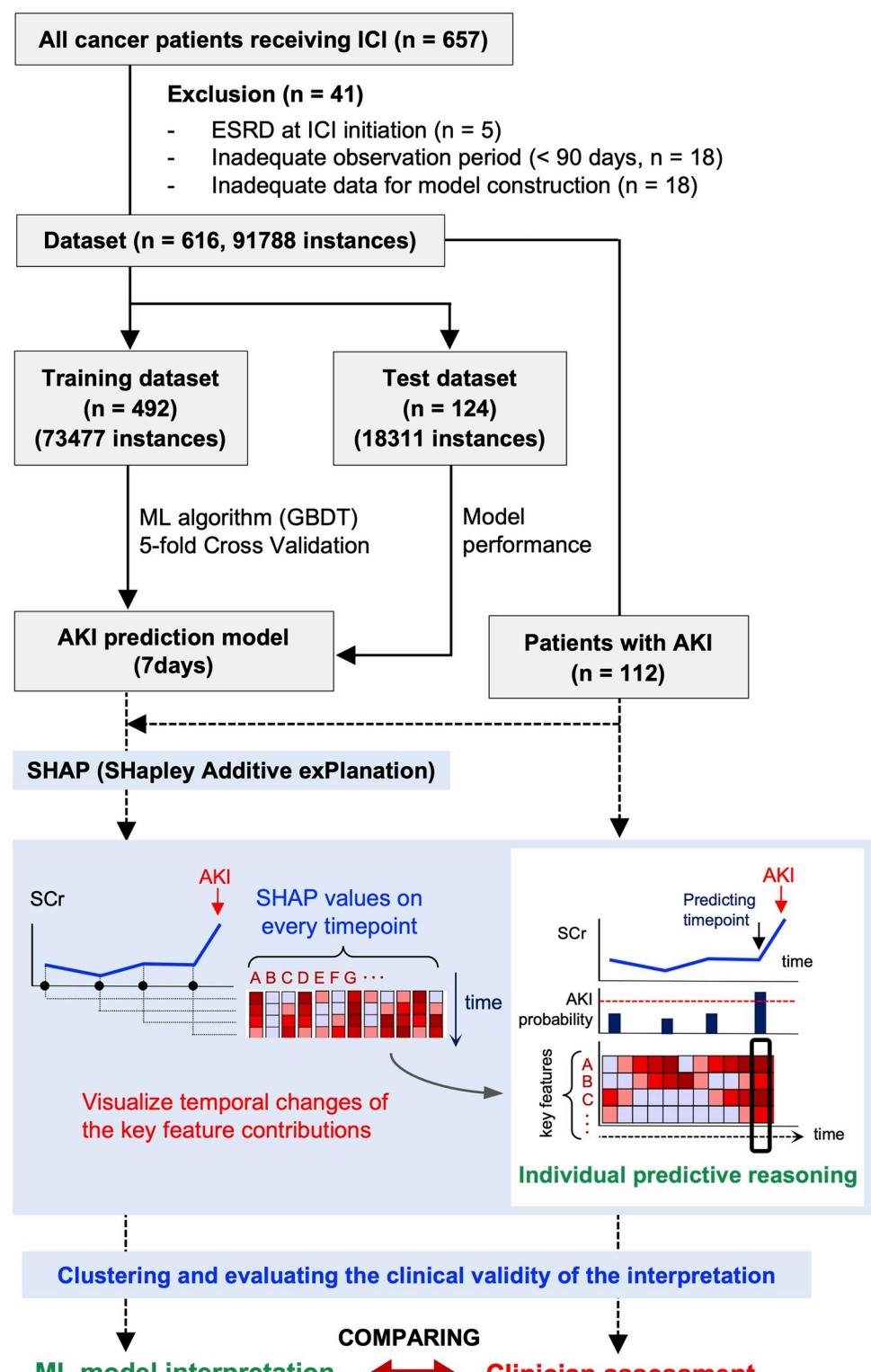

**Fig 1. Analysis overview.** Among the 657 ICI-treated patients, those who had end-stage renal disease (ESRD) before ICI initiation and those with missing data or inadequate periods for model construction were excluded from this study. The entire dataset was split into test (20%) and training datasets on a per-patient basis, and hyperparameter tuning of the model was performed on the training dataset (5-fold cross-validation). The contribution of key features to AKI development at each prediction time point was quantified based on SHAP values and visualized using the

heatmap. The SHAP value of each feature takes a positive or negative value as a vector of contributions, with the magnitude of the absolute value representing the degree of influence on the prediction outcome. The trend of SHAP values at the time point when the model predicts AKI (bold black box) indicates the combination of key features and their contributions (individual predictive reasoning) crucial for predicting AKI in that patient. AKI, acute kidney injury; ICI, immune checkpoint inhibitors; ESRD, end-stage renal disease; ML, machine learning; GBDT, Gradient Boosting Decision Tree; SHAP, SHapley Additive exPlanations; SCr, serum creatinine.

## Ethical statement and informed consent

The dataset was generated and reviewed based on the clinical information obtained from the EMR of our institution. This study was conducted using data obtained only during medical practice, according to the principles of the Declaration of Helsinki. Per Japanese laws and regulations, informed consent was obtained on an opt-out basis. All explanations of the study and expressions of consent were assured to be conducted in a written format, guaranteeing that participants received comprehensive information and their consent or dissent was appropriately recorded. This method aligns with the approval granted by the Ethical Review Board of Kyoto University, acknowledging it as a valid form of consent for this type of research. We ensured ethical compliance by publicly providing detailed information about the study, including its purpose, the nature of the data used, and the rights of participants to withdraw, on the Kyoto University Hospital website (https://www.kuhp.kyoto-u.ac.jp/outline/research-disclosure.html). The option for participants was made clear and accessible, thus preserving their autonomy. The Ethical Review Board of Kyoto University approved the study (Approval Number R1498), recognizing its adequacy for the nature of this retrospective analysis. The period of data access and analysis for this study was from March 2022 to August 2022. In collecting the data, the authors did not access any data that could identify individual participants.

## Results

### Model performance and visualizing individual predictive reasoning

Among the 616 patients, 112 (18.2%) had at least one AKI episode after initiation of ICI therapy. The clinical characteristics of the patients are presented in Table 1. The generalization performance of the model estimated based on the test data had an area under the receiver operating characteristic curve of 0.880, similar to that of the pre-existing models [22–29] (Fig 2a). Performance comparisons with other ML models are summarized in S1 Table in S1 File. The SHAP values of the key features that contributed to the prediction of AKI are presented in Fig 2b and 2c. Two examples of predictive reasoning in patients with AKI are presented in Fig 2d. Considering that the contributing factors of AKI vary across patients (Fig 2e), individual differences in predictive reasoning may reflect individual differences in clinical backgrounds related to the development of AKI.

### Clustering patients with AKI using predictive reasoning

A total of 112 patients with AKI were categorized into four clusters based on predictive reasoning immediately before the first episode of AKI using unsupervised clustering (Fig 3a, Table 2), compared with annotation independently reviewed by the three nephrologists [24]. The number of clusters was determined as the number of visually valid clusters indicated on the dendrogram produced by the hierarchical clustering. Based on their descriptions, the strongest contributive risk factors for AKI development were assigned six labels for each patient: "Hypovolemia," "Cancer Cachexia," "Infection," "Drug-related," "Obstruction," and "Others." (S3 Method in S1 File). The number of these labels was counted in each cluster to

**Table 1. Baseline characteristics of patients undergoing immune checkpoint inhibitor therapy.**

| | All patients (n = 616) | | | |
|---|---|---|---|---|
| | | With AKI (n = 112) | Without AKI (n = 504) | p-value |
| Age [n (%)] | | | | |
| 20–39 years | 17 (3) | 3 (3) | 14 (3) | 0.954 |
| 40–59 years | 107 (17) | 21 (19) | 86 (17) | 0.773 |
| 60–79 years | 415 (67) | 80 (71) | 335 (66) | 0.367 |
| > 80 years | 77 (13) | 8 (7) | 69 (14) | 0.082 |
| Male [n] / Female [n] | 415 / 201 | 82 / 30 | 333 / 171 | 0.178 |
| Malignancy types [n (%)] | | | | |
| Gastrointestinal | 74 (12) | 10 (9) | 64 (13) | 0.342 |
| Lung | 333 (54) | 45 (40) | 288 (57) | 0.001 * |
| Urologic | 72 (12) | 23 (21) | 49 (10) | 0.002 * |
| Skin | 78 (13) | 28 (25) | 50 (10) | < 0.001 * |
| Other | 59 (9) | 6 (5) | 53 (11) | 0.133 |
| ICI types [n (%)] | | | | |
| PD-1 antibody | 559 (91) | 103 (92) | 453 (90) | 0.620 |
| PD-L1 antibody | 75 (12) | 11 (10) | 64 (13) | 0.495 |
| CTLA-4 antibody | 43 (7) | 12 (11) | 31 (6) | 0.131 |
| Combination therapy | 22 (4) | 4 (4) | 18 (4) | 1.000 |
| Baseline SCr [mg/dL, median (IQR)] | 0.79 (0.66–0.95) | 0.90 (0.67–1.11) | 0.82 (0.66–0.92) | < 0.001 * |
| PPI administration [n (%)] | 152 (25) | 33 (29) | 119 (24) | 0.239 |
| NSAID administration [n (%)] | 66 (11) | 12 (11) | 54 (11) | 1.000 |

All data are presented as medians (interquartile range, IQR) or means (standard deviation, SD), as appropriate for nonparametric or parametric variables, respectively. Patients with ESRD at the initiation of ICI (n = 5), patients without data on renal function after ICI (n = 18), and patients whose follow-up was censored < 3 months after initiation of ICI (n = 18) were excluded from the analysis. ICIs included anti-PD-1, anti-PD-L1, and anti-CTLA-4 antibodies, while some patients received combination therapy with anti-PD-1 and anti-CTLA-4 antibodies. Comparisons of categorical variables were made using the Chi-square test or Fisher's exact probability test. ICI, immune checkpoint inhibitors; AKI, acute kidney injury; PD-1, Programmed cell death 1; PD-L1, Programmed death-ligand 1; CTLA-4, Cytotoxic T-lymphocyte-associated antigen; SCr, serum creatinine; PPI, proton pump inhibitors; NSAIDs, nonsteroidal anti-inflammatory drugs; ESRD, end-stage renal disease; IQR, interquartile range; SD, standard deviation.

determine the most dominant contributive risk factor. Although the proportions of each label did not differ significantly among the clusters, each cluster had distinct patterns of contributing risk factors for the development of AKI (Fig 3b). While there was a clear trend in the label distribution within each cluster, only clusters 3 and 4 showed statistically significant differences. The most dominant labels in each cluster were as follows: cluster 1, "Hypovolemia"; cluster 2, "Drug-related"; cluster 3, "Drug-related"; and cluster 4, "Cancer Cachexia." In addition, clusters 2 and 3 were annotated as "Drug-related," including IrAE, while each cluster indicated different patient backgrounds (S2 Table in S1 File). These results suggested that patients categorized by predictive reasoning likely have different clinical backgrounds regarding AKI development between the clusters.

To further elucidate patient clustering by SHAP, we constructed a two-dimensional plot (dependence plot), which represents the correlation between the feature values and their SHAP values in the week before AKI development among 112 patients with AKI (Fig 3c). For example, cluster 4, which was characterized by high SHAP values for C-reactive protein (CRP) and lactate dehydrogenase (LDH), dietary intake, and diuretic use, demonstrated high CRP and LDH levels and poor dietary intake, including diuretic use in one out of three cases, which strongly contributed to AKI prediction (Fig 3a). Generally, high CRP and LDH levels and

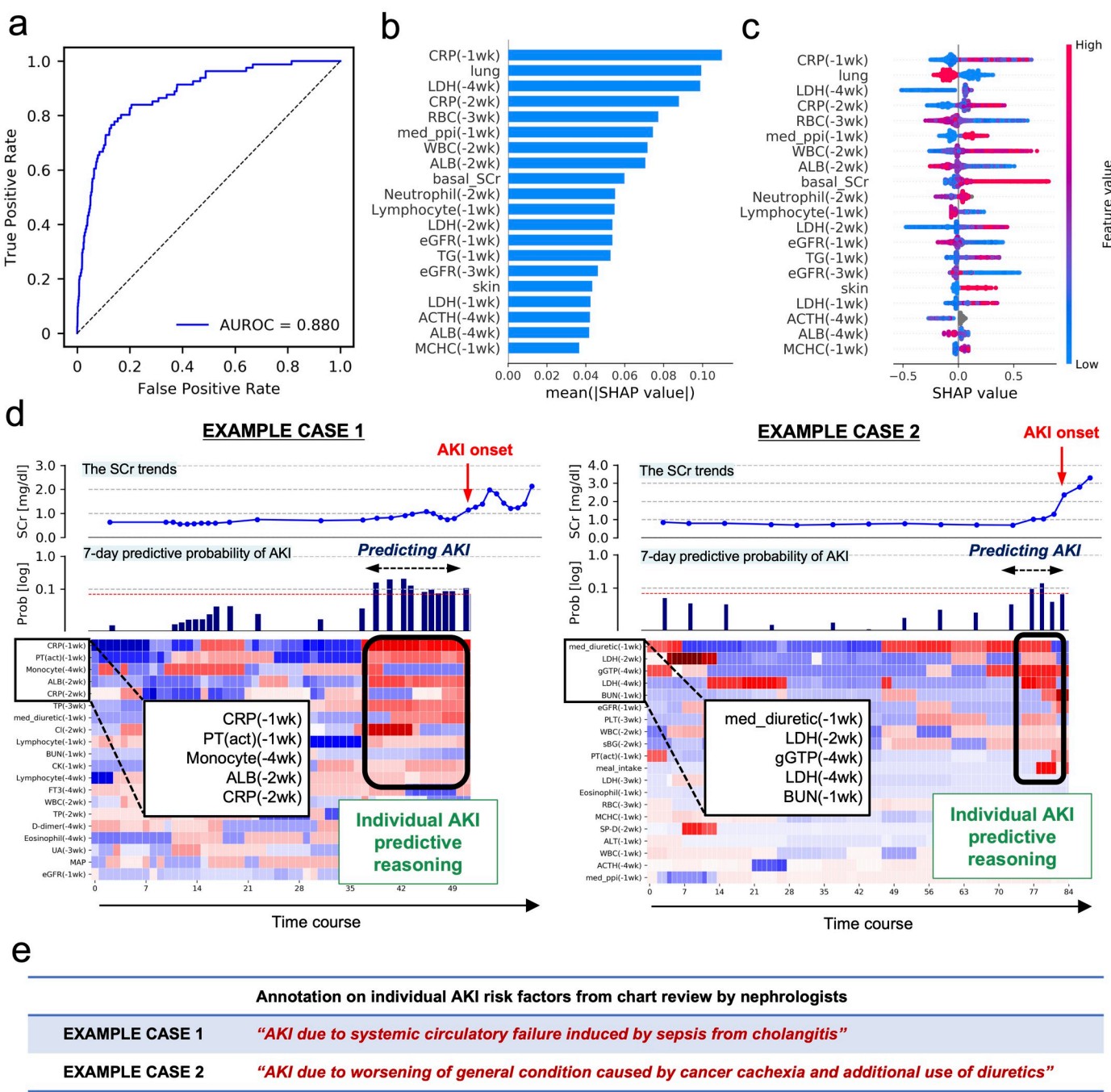

**Fig 2. Model performance and visualizing individual predictive reasoning.** (a) Performance of the model. The general performance is evaluated based on the area under the ROC curve. (b, c) Features indicating higher overall SHAP. Features with higher average contributions for all patients are shown. Positive and negative contributions to predicted AKI development are characterized by positive and negative SHAP values, respectively, with red and blue representing the magnitude of respective feature values. (d) Examples of individual predictive reasoning. The graph of SCr value-predicted probabilities of AKI development within 7 days, and the heatmap of SHAP values for the key features are represented on the same timeline. The red dotted line indicates the threshold value for 0.25 in the precision probabilities, and the predicted probabilities above the line are regarded as positive predictions (S5 Fig in S1 File). The bold black-boxed area, at time points with elevated predictive probability, represents the key features and their contribution to the prediction of AKI for that individual. These two examples demonstrate different heatmap patterns of SHAP, suggesting the difference in predictive reasoning. (e) Examples of nephrologists' chart reviews. The annotations of nephrologists for contributing factors to AKI development for the two cases with predictive reasoning are shown above. ROC, receiver operating characteristic curve; SHAP, SHapley Additive exPlanations; AKI, acute kidney injury; SCr, serum creatinine; IrAEs, immune-related adverse events.

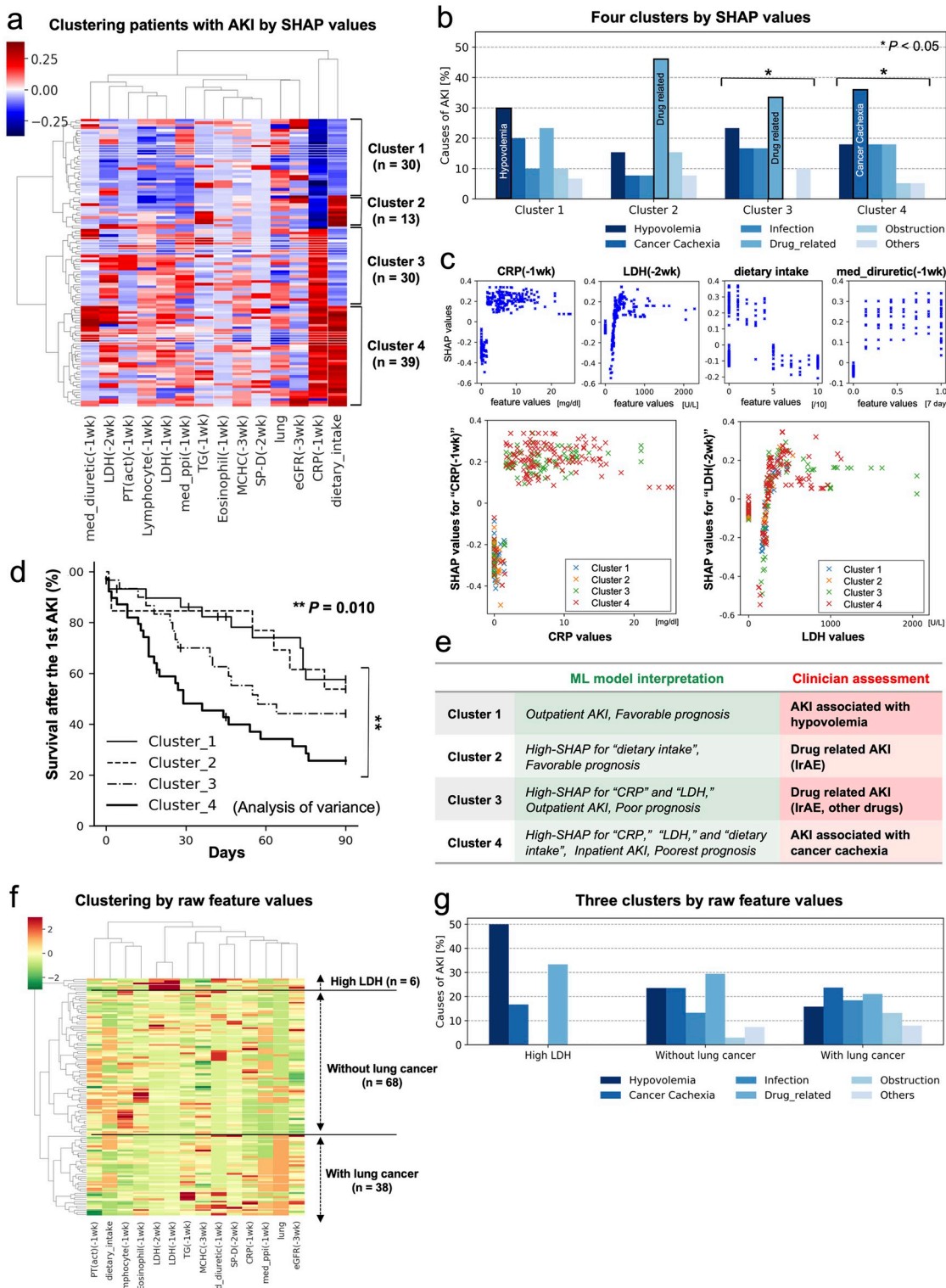

**Fig 3.** (a) Patient clustering by SHAP values. Overall, 112 patients with AKI were categorized into four clusters with ML-based unsupervised clustering by SHAP values. The number of clusters was determined as the number of visually valid clusters indicated on the dendrogram. (b) Distribution of annotations for causes of AKI in the four clusters. The distribution of annotation labels was calculated cluster-wise. (c) Dependence plot of key features. Each point in the scatterplot reveals the correlation between the values of key features and SHAP in the last week prior to each AKI development among 112 patients. The plots in the features of CRP and

LDH are color-coded by cluster. (d) Survival analysis after AKI. Kaplan—Meier curves of 90-day survival after the first AKI in each cluster. Analysis of variance reveals significant differences in survival rates among the four clusters, with Cluster 4 having the poorest prognosis. (e) Interpretation of the AI model and clinician assessment. The interpretation of the AI-based model is indicated by patient clustering based on predictive reasoning and prognostic variance. Clinician assessment is indicated by the reviews of nephrologists. (f) Patient clustering by raw feature values. Clustering by raw values of the key features, excluding SHAP weighting, categorizes the same patients with AKI into three clusters. (g) Distribution of annotations for causes of AKI in the three clusters. As in (b), six labels were aggregated for each cluster with the contributing factors for AKI. Clustering by raw values of the key features does not provide meaningful patient clustering reflecting the clinical background of AKI. SHAP, SHapley Additive exPlanations; AKI, acute kidney injury; CRP, C-reactive protein; LDH, Lactate Dehydrogenase; med_diuretic(-1wk), medication of diuretics within the last week.

poor dietary intake are associated with organ damage and dehydration, which can be causes of AKI in advanced cancers [4]. According to the chart review by nephrologists, certain patients with AKI in cluster 4 had cancer cachexia in the terminal phase, while some developed diuretic-induced AKI. Based on these findings, the AKI predictive reasoning in cluster 4 can be interpreted as "patients with terminal cancer and cachexia who developed AKI due to worsening conditions or diuretic use, high CRP and LDH levels, and poor dietary intake." In

**Table 2. Clinical characteristics of patients with acute kidney injury in each cluster.**

|  | Cluster 1 | Cluster 2 | Cluster 3 | Cluster 4 | p-value |
|---|---|---|---|---|---|
| Number of patients [n] | 30 | 13 | 30 | 39 |  |
| Age [n (%)] |  |  |  |  |  |
| 20–39 years | 0 (0) | 1 (8) | 0 (0) | 2 (5) | 0.203 |
| 40–59 years | 6 (20) | 2 (15) | 8 (27) | 5 (13) | 0.590 |
| 60–79 years | 21 (70) | 9 (69) | 20 (67) | 30 (77) | 0.720 |
| > 80 years | 3 (10) | 1 (8) | 2 (7) | 2 (5) | 0.953 |
| Male [n] / Female [n] | 22 / 8 | 7 / 6 | 19 / 11 | 34 / 5 | < 0.05 * |
| Malignancy types [n (%)] |  |  |  |  |  |
| Gastrointestinal | 4 (13) | 1 (8) | 4 (13) | 1 (2) | 0.262 |
| Lung | 10 (33) | 5 (38) | 14 (47) | 16 (41) | 0.780 |
| Urologic | 8 (27) | 5 (38) | 3 (10) | 7 (18) | 0.108 |
| Skin | 7 (23) | 1 (8) | 8 (27) | 12 (31) | 0.492 |
| Other | 1 (3) | 1 (8) | 1 (3) | 3 (8) | 0.730 |
| Baseline SCr [mg/dL, median (IQR)] | 0.91 (0.77–1.12) | 0.75 (0.62–1.32) | 0.85 (0.62–0.99) | 0.92 (0.73–1.18) | 0.543 |
| AKI stage on first episode of AKI [n (%)] |  |  |  |  |  |
| Stage 1 | 19 (63) | 8 (62) | 27 (90) | 26 (67) | < 0.05 * |
| Stage 2 | 6 (20) | 3 (23) | 2 (7) | 8 (20) | 0.322 |
| Stage 3 or require RRT | 5 (17) | 2 (15) | 1 (3) | 5 (13) | 0.343 |
| Ratio of inpatient AKI [n (%)] | 3 (10) | 6 (46) | 7 (23) | 36 (92) | < 0.05 * |
| Primary cause of AKI [n (%)] |  |  |  |  |  |
| Hypovolemia | 9 (30) | 2 (15) | 7 (23) | 7 (18) | 0.634 |
| Cancer Cachexia | 6 (20) | 1 (8) | 5 (17) | 14 (36) | 0.125 |
| Infection | 3 (10) | 1 (8) | 5 (17) | 7 (18) | 0.765 |
| Drug-related | 7 (23) | 6 (46) | 10 (33) | 7 (18) | 0.174 |
| Obstruction | 3 (10) | 2 (15) | 0 (0) | 2 (5) | 0.142 |
| Others | 2 (7) | 1 (8) | 3 (10) | 2 (5) | 0.953 |

All data are presented as medians (interquartile range, IQR) or means (standard deviation, SD), as appropriate for nonparametric or parametric variables, respectively. Comparisons of categorical variables and means among clusters are made using Fisher's exact probability test and the Kruskal—Wallis test, respectively. AKI, acute kidney injury; ICI, immune checkpoint inhibitors; SCr, serum creatinine; RRT, renal replacement therapy; IQR, interquartile range; SD, standard deviation.

addition, when the dependence plots of CRP and LDH were color-coded by cluster, higher values of the features and SHAP were frequently observed in clusters 3 and 4 (Fig 3c). Furthermore, since the clustering with AKI predictive reasoning captured the distinct clinical characteristics of cancer patients, we speculated that patient clustering by SHAP may capture prognostic differences in advanced cancers. Therefore, the 90-day survival rate of 112 patients with the first occurrence of AKI was analyzed, and it was discovered that significant prognostic differences existed between the four clusters (Fig 3d). Notably, cluster 4 had the poorest prognosis. These findings suggest that the predictive reasoning for AKI can recognize prognostic variances after AKI, supporting the clinical validity of patient clustering by SHAP (Fig 3e).

To confirm the necessity of SHAP in clinical interpretation, the same patients were clustered by the raw values for the same key features and divided into three clusters (Fig 3f). The results revealed that, in contrast to SHAP clustering, there were no distinguishing characteristics in the causes of AKI between the clusters, and each cluster did not reflect the contributing risk factors for AKI development (Fig 3g).

Among the patients in clusters 2 and 3, only a few cases of suspected ICI or IrAE involvement were confirmed on renal biopsy. A detailed chart review revealed that many cases were not biopsied for AKI diagnosis after discussions among the attending physician, patient, and their family; consideration of the general condition of the procedure; the prognosis of the patient; and the risk of fatal complications.

## Discussion

Herein, we have shown that the clustering approach using SHAP values in ML-based AKI prediction models offers a novel perspective in assessing the etiology of each episode of AKI in patients undergoing ICI therapy. Patient clustering based on time-series SHAP values for AKI prediction enables clinicians to interpret predictive reasoning that reflects the underlying causes of AKI individually. This indicates that we can infer factors critical for AKI development on a case specific basis by focusing on the temporal changes and patterns in each SHAP value in the ML model, which continuously predicts AKI. Therefore, our approach seems appropriate for estimating the most critical causes of AKI in cancer patients receiving systemic therapy, including ICI therapy, with diverse and complicated AKI risks. The features predicted as particularly essential variables in our model were consistent with the findings of previous studies using multivariate analyses [11–15]. PPIs, which have been associated with the development of AKI in several observational studies [12, 13, 30], were also identified as a key feature in our prediction model. In addition, although not at the top of the list, diuretics, NSAIDs, and baseline renal function features were also identified as key risk factors by the model, as shown in the dependence plot [17] (Fig 3c, S4 Fig in S1 File). Although the dependence plot did not indicate a causal relationship, the prediction model regarded these key features as crucial for predicting AKI.

However, our method identified individual differences in the underlying backgrounds of AKI that could not have been deduced by conventional methods. As indicated by the varying distribution of clinician annotations (Fig 3b), the patient clusters classified based on the predicted key features had different AKI development backgrounds; for example, cluster 4 was interpreted to have cancer cachexia as the primary contributing factor to AKI development, whereas clusters 2 and 3 suggested the contribution of drugs, including ICI or IrAE. Patients in cluster 4 were characterized by high CRP and LDH levels and the use of diuretics and had the poorest prognosis after the development of AKI (Fig 3c and 3d). These predictive findings reflect the development of AKI due to cancer cachexia. Cluster 3 had more cases of higher CRP levels, persistent inflammation due to IrAE, infections, and end-stage cancer, with many

patients receiving outpatient follow-ups. The high SHAP trend of CRP in cluster 3 was considered reflective of these conditions. In contrast, cluster 2 had relatively more cases of poor dietary intake that required hospitalization and fever. The high SHAP trend of dietary intake in cluster 2 was considered to reflect these conditions. Most patients with drug-related AKI in clusters 2 and 3 developed extra-renal IrAEs before AKI [15] (S2 Table in S1 File). In addition, significant prognostic differences were noted between the clusters according to the predictive reasoning, although no variable for survival was provided for model training. This indicates that the predictive reasoning of the AI model is not solely derived from a combination of laboratory values and medications. Although some studies have discussed the prognostic relevance of AKI in ICI-treated patients [11, 13, 14], our study suggested that prognostic differences after AKI were relevant regarding the differences in predicted factors of AKI development.

In several AI-based prediction models, SHAP has been widely used to predict risk factors for various outcomes, including AKI [31–34]. However, although it is possible to infer predicted characteristics that demonstrate measurable correlations with SHAP values, it has not been feasible to determine their clinical significance in individual patients. This is partly because the correlation between an individual input feature and its contribution does not fully explain the pathophysiology of complicated outcomes. Furthermore, although many features with nonlinear relationships with SHAP values contribute to the prediction of AKI (S4 Fig in S1 File), comprehending the clinical importance of each feature with a nonlinear contribution is challenging. To the best of our knowledge, no study has attempted to clinically interpret the meaning of contributing factors as individual risk factors in each patient. We found that the combination of contributing factors, including nonlinear contributions, constitutes predictive reasoning in AI models representing the time-varying AKI risks. This method allowed us to clinically interpret the underlying background behind individualized prediction of AKI observed in different time series for the first time.

We believe that our study is significant because it reveals underlying causes in individual patients with AKI in ICI therapy, which cannot be obtained by conventional approaches, and provides predictive reasoning with clinically valid interpretability. However, the implications of our study go beyond simply allowing individualized assessment of AKI during ICI therapy. Cancer patients typically develop AKI owing to complex risk factors arising from various medications or complications. Therefore, predicting AKI development by monitoring a single laboratory result or medication considered as critical factors is often difficult. Similar to investigating the significant contributive features by SHAP analysis, determining the most critical factor for AKI among the multilayered AKI risk factors is a process that clinicians implement to select patients at high risk of AKI and assess their risks. Clinicians usually follow thought processes such as "the probability of AKI onset increases when additional risk factors such as infections and diuretics (triggers) are added to the background of cancer cachexia (underlying risks)." When interpreting the combination of underlying clinical backgrounds and additional stratified risks that lead to AKI development, analyzing the individual AI models' predictive reasoning can be a valuable approach to explore the most critical AKI risks, which are challenging to understand using routine medical data [35]. In the future, this approach will help effectively determine the appropriate assessment and intervention for patients with complicated AKI risks (S5 Fig in S1 File) [36]. Further analyses applying a similar approach to patients receiving other chemotherapy may capture other characteristic predicting reasoning models specific to the causative agent and disease state. Furthermore, this model can be applied to predict AKI and other outcomes in various other fields that need such individualized prediction.

This study had several limitations. First, this model was developed at a single center; hence, multicenter studies are needed for external validation. Second, due to the nature of ICI

therapy, the difference in data availability may have affected the prediction accuracy and the contribution of the features (S6 Fig in S1 File). Therefore, designing equal time-series features, devising missing interpolations, and selecting the population may resolve this problem. Third, information on image findings and surgery, which may be necessary for specific AKI prediction (e.g., obstructive AKI), were not included as features in the present model. Therefore, adding such information in future studies can further improve the performance and interpretability of the model. Furthermore, the validity of the clinical interpretation was assessed by reviews conducted by nephrologists; however, information may have been missed in the retrospective chart reviews. Finally, although this was a retrospective analysis by design, future prospective studies are expected to clarify the benefits of patient clustering by predictive reasoning, which can aid clinicians' decisions and patient outcomes by prospectively predicting new patients with AKI.

In conclusion, the study findings are significant as this study is the first to demonstrate a novel approach for interpreting ML models by patient clustering using individual predictive reasoning patterns and has the potential to accelerate future medical applications of AI. We expect our approach to be widely applied to explainable AI in various medical fields, including renal diseases.

## Supporting information

**S1 File. Contains all the supporting files.**
(PDF)

## Acknowledgments

We thank Tomohiro Kuroda and the Division of Medical Informatics and Administration Planning, Kyoto University Hospital, for the EMR data extraction and management. We also like to thank Editage (www.editage.com) for English language editing.

## Author Contributions

**Conceptualization:** Minoru Sakuragi.

**Data curation:** Minoru Sakuragi, Eiichiro Uchino, Noriaki Sato.

**Formal analysis:** Minoru Sakuragi.

**Funding acquisition:** Yasushi Okuno.

**Investigation:** Minoru Sakuragi.

**Methodology:** Minoru Sakuragi, Eiichiro Uchino, Noriaki Sato.

**Project administration:** Motoko Yanagita, Yasushi Okuno.

**Resources:** Motoko Yanagita, Yasushi Okuno.

**Software:** Minoru Sakuragi, Eiichiro Uchino, Ryosuke Kojima.

**Supervision:** Motoko Yanagita, Yasushi Okuno.

**Validation:** Takeshi Matsubara, Akihiko Ueda, Yohei Mineharu.

**Visualization:** Minoru Sakuragi.

**Writing – original draft:** Minoru Sakuragi.

**Writing – review & editing:** Eiichiro Uchino, Noriaki Sato, Takeshi Matsubara, Akihiko Ueda, Yohei Mineharu, Motoko Yanagita, Yasushi Okuno.

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
