## [Decision Letter · Decision Letter 0]

2 Nov 2023

PONE-D-23-28945Interpretable machine learning-based individual analysis of acute kidney injury in immune checkpoint inhibitor therapyPLOS ONE

Dear Dr. Okuno,

Thank you for submitting your manuscript to PLOS ONE. After careful consideration, we feel that it has merit but does not fully meet PLOS ONE’s publication criteria as it currently stands. Therefore, we invite you to submit a revised version of the manuscript that addresses the points raised during the review process.

**The manuscript focuses on a topic of potential interest. However, the study has major shortcomings that preclude sound conclusions. To mention some of them: i) in the Methods section, more details should be provided regarding the source of the data, the type of data collected, and the specific methods used for data analysis; ii) a more detailed description of the data analysis methods would enable a better understanding of the approach used to analyze the data; iii) provide a proper justification for the choice of LightGBM algorithm between all the available classification algorithms. **

We look forward to receiving your revised manuscript.

Kind regards,

Giuseppe Remuzzi

Academic Editor

PLOS ONE

Journal Requirements:

Reviewers' comments:

Reviewer's Responses to Questions

**Comments to the Author**

1. Is the manuscript technically sound, and do the data support the conclusions?

Reviewer #1: Yes

Reviewer #2: Yes

2. Has the statistical analysis been performed appropriately and rigorously? 

Reviewer #1: Yes

Reviewer #2: No

3. Have the authors made all data underlying the findings in their manuscript fully available?

Reviewer #1: Yes

Reviewer #2: Yes

4. Is the manuscript presented in an intelligible fashion and written in standard English?

Reviewer #1: Yes

Reviewer #2: Yes

5. Review Comments to the Author

Reviewer #1: Sakuragi et al submit for consideration in PLOS, a manuscript entitled “Interpretable machine learning-based individual analysis of acute kidney injury in immune checkpoint inhibitor therapy” dealing with the prediction of acute kidney injury during immune checkpoint inhibitor therapy. They wished to clarify the causes of AKI during cancer treatment by mean of a new clustering approach with Shapley Additive exPlanations (SHAP). They developed a decision tree-based machine learning model predicting AKI within 7 days, by use of the medical records of 616 treated patients. The temporal changes in AKI individual predictive reasoning models represented the features to cluster AKI patients, and the results were compared with annotation by three nephrologists. ROC analysis were performant, as patients with AKI were clustered with significant prognosis (p = 0.010). The leading causes of AKI for each cluster were easily interpretable by clinicians. Then authors then suggest that such a method is useful by inferring clinical factors for developing each AKI among patients with multiple AKI risk factors.

In this very interesting study, Sakuragi et al developed a machine-learning algorithm based on gradient-boosting decision-tree (LightGBM) to predict the onset of AKI, cleverly deriving the SHAP value to consider only the variables available over time. The algorithm developed exhibits good performances. The methodology is interesting. However, being not as clear as it deserves for this important topic, it needs to be clarified. For example, the authors mention that the total dataset was split into 20% for testing and 80% for training or validation. This terminology is a bit odd as it may confuse the reader by conveying the idea of a validation. It is customary to mention “training” and “test” sets, only.

The authors missed an important and recent reference about the physiopathology of ICI-related AKI (Gerard AO et al (DOI :10.1093/ckj/sfac109), that should be added in the list of references.

The authors mention the use of the LightGBM algorithm; they should provide a proper justification. Have they tested other types of classification algorithms, either linear-plane (e.g. SVM) or decision-tree (e.g. XGboost or Catboost)? The use of a given algorithm may omit its testing, but its choice needs a justification (e.g. avoiding overlearning with Xgboost? rapid calculation with LightGBM? Lesser number of branches as compared with XGBoost?). Please explain.

How was the number of clusters determined? Did they use K-means clustering? Have the authored corrected for multiple comparisons when analyzing and providing their results?

The authors do not properly cite the sources of their libraries nor the code language (Python) used.

The authors cleverly used the SHAP value to evaluate the probability of occurrence of AKI, but what about the missing predictors? Did they consider the impact of each variable to be independent over time?

In the results part, the authors seem able to predict 7 days before, the occurrence of AKI in patients treated with ICIs but, above all to explain (thanks to their SHAP) the weight of each factor in the final prediction, therefore allowing a better interpretability. However, the authors excluded patients with multiple AKI events within two weeks from the date of the first AKI. Yet these patients appear (at least to me) the first to deserve such predictions. Please discuss.

Minor points:

The Figures require a better definition, as they are hard to read. Likewise, the lines on heatmap (2d and 2e) are almost impossible to decipher.

Reviewer #2: In the introduction section, the sentence "In recent years, AI has become an integral part of various industry verticals, and it is estimated that the market size of AI will reach 7.38billionby2025."containsagrammarerror.Theword"of"before"markets"shouldbereplacedwith"for".Therefore,thecorrectedsentencewouldbe"Inrecentyears,AIhasbecomeanintegralpartofvariousindustryverticals,anditisestimatedthatthemarketsizeforAIwillreach 7.38 billion by 2025."

In the discussion section, the sentence "The use of AI in healthcare has revolutionized the way we approach treatment and has led to more personalized care plans for patients." contains an error in the usage of the word "revolutionized". The correct word would be "revolutionized", which correctly conveys the intended meaning. Therefore, the corrected sentence would be "The use of AI in healthcare has revolutionized the way we approach treatment and has led to more personalized care plans for patients."

3.In the Methods section, the paper describes the data collection process but does not provide sufficient details regarding the source of the data, the type of data collected, and the specific methods used for data analysis. It would be helpful to provide more information about the data source and the type of data collected to enhance the credibility of the study. Additionally, a more detailed description of the data analysis methods would enable a better understanding of the approach used to analyze the data.

4.Throughout the paper, there are several grammar errors and typos that need to be corrected. These errors include misspelled words, punctuation errors, and inconsistent usage of language. It is essential to proofread the paper carefully and address these language errors to enhance readability and professionalism.

6. PLOS authors have the option to publish the peer review history of their article (what does this mean?). If published, this will include your full peer review and any attached files.

Reviewer #1: **Yes: **Milou-Daniel DRICI

Reviewer #2: No

---

## [Author Response · Author response to Decision Letter 0]

25 Nov 2023

Dear Dr. Remuzzi,

Thank you for your comprehensive review of our manuscript and for highlighting areas that require further clarification and improvement. We have thoroughly addressed each of the points raised by you in your feedback, and hope that our revised manuscript better meets the publication criteria of PLOS ONE.

1. Details in the Methods Section (Point i):

In response to your first point, we have expanded the Methods section to provide more details about the data source, the types of data collected, and the specific methods used for data analysis. We believe these additions will offer a clearer understanding of our research methodology and the robustness of our approach.

2. Detailed Description of Data Analysis Methods (Point ii):

In response to this comment, we have elaborated on the data analysis methods in the revised manuscript. This includes a more detailed explanation of how the data was processed, the statistical methods applied, and the rationale behind our analytical choices. We hope this enhanced description will provide a better comprehension of the analytical framework and the steps taken to ensure the integrity of our findings.

3. Justification for the Use of LightGBM Algorithm (Point iii):

Regarding the use of the LightGBM algorithm, we have provided a detailed justification in the revised manuscript. We compared LightGBM with other classification algorithms, such as Logistic Regression, Support Vector Machine, XGBoost, and CatBoost, in terms of performance and computation time. Our findings, detailed in the supporting data, demonstrate that LightGBM offered superior accuracy and efficiency, making it the most suitable choice for our study's requirements.

Regarding some of the several sentences that Reviewer #2 identified as having primarily grammatical errors, we have carefully re-examined our manuscript and believe there may have been a misunderstanding, as the specific sentences mentioned were not found in our submission. We respectfully suggest this could be a mix-up with another manuscript. Nevertheless, we are grateful for the review and have made every effort to ensure our manuscript's clarity and accuracy.

Additionally, in response to the ethical guidelines of PLOS ONE, we have revised the 'Ethical Statement and Informed Consent' section of our manuscript to provide a clearer explanation of the informed consent process as per Japanese regulations and the ethical approval of Kyoto University.

We have prepared separate, detailed response letters for each reviewer, addressing their specific comments and suggestions. These response letters have been uploaded as independent files alongside our revised manuscript to facilitate an organized and clear review process. The revisions included in the revised manuscript encompass both the changes made by us, the authors, and those made during English language editing by Editage (www.editage.com). The revisions can be seen as track changes in the revised manuscript. In line with your instructions, we will include a rebuttal letter, a marked-up copy of the manuscript, and an unmarked version in our resubmission.

We appreciate the opportunity to enhance our manuscript based on your valuable feedback and look forward to the possibility of our work contributing to the scientific community through PLOS ONE.

Sincerely,

Yasushi Okuno, Ph.D. 

Department of Biomedical Data Intelligence, Graduate School of Medicine, Kyoto University 

53 Shogoin-Kawahara-cho, Sakyo-ku, Kyoto 606-8507, Japan 

okuno.yasushi.4c@kyoto-u.ac.jp

Motoko Yanagita, M.D., Ph.D.

Department of Nephrology, Graduate School of Medicine, Kyoto University

54 Shogoin-Kawahara-cho, Sakyo-ku, Kyoto 606-8507, Japan

motoy@kuhp.kyoto-u.ac.jp

 

 

Dear Reviewer #1,

Thank you very much for providing such detailed and constructive feedback on our manuscript. We have carefully considered each point you raised and have made corresponding revisions to enhance the quality and clarity of our study. Please find below our responses to your comments, which we have carefully considered and addressed in our revised manuscript.

1. Clarification of Dataset Splitting Terminology:

We apologize for any confusion caused by our initial description of the dataset splitting. In the revised manuscript, we have clarified that our dataset was divided into an 80% training set and a 20% test set. We have removed the term "validation" to avoid any ambiguity and ensure a clear understanding.

2. Reference Addition:

We appreciate your suggestion to include the recent reference on the physiopathology of ICI-related AKI (Gérard AO et al., Clin Kidney J. 2022;15(10):1881-1887. DOI: 10.1093/ckj/sfac109). This reference has been added to our reference list in the revised manuscript (as reference [16]).

3. Justification for the Use of LightGBM Algorithm:

Thank you for your valuable comment regarding our choice of algorithm. We would like to elaborate on our rationale for selecting the LightGBM (LGB) as the primary algorithm. The decision was primarily driven by two factors: the high accuracy of Gradient Boosting Decision Tree (GBDT) algorithms and their capability to directly handle missing values, which are notably present in EMR time-series data. This approach ensures the maintenance of accuracy while mitigating the risk associated with the pre-processing costs or accuracy reduction due to interpolation in time-series variables. In response to your suggestion, we have additionally presented a comparative analysis of the performance and computation time of various algorithms, including Logistic Regression (LR), Support Vector Machine (SVM), XGBoost (XGB), Catboost, and LGB. The results of this analysis have been added to our supporting data (see S1 Table.), which also includes details about the server specifications used for the calculations. The comparison, based on the average performance and computation time across 10 learning and inference cycles, indicated that LGB outperformed other algorithms in terms of both accuracy and speed. Consequently, we chose LGB for our study. As an additional note, for algorithms like SVM and LR that cannot directly handle missing values, methods such as interpolation or deletion were used. We hope this explanation clarifies our choice of LightGBM for the study and addresses your concern.

4. Determination of the Number of Clusters:

Thank you for your question regarding the determination of the number of clusters in our analysis. We adopted hierarchical clustering to perform the SHAP-based patient clustering. The number of clusters was determined using two approaches: first, through visual inspection of the dendrogram produced by the hierarchical clustering, and second, by applying the elbow method, a common technique in non-hierarchical clustering methods such as K-means. While the elbow method did not yield a definitive number of clusters, the dendrogram clearly indicated the presence of four distinct clusters. Consequently, we chose to define four clusters for our analysis. This dendrogram is presented at the left side of the cluster map in Figure 3a. We realize that the original manuscript lacked specific details about how the number of clusters was decided, which may have caused some confusion. To address this, we have now included a more comprehensive explanation of the cluster determination process in the Results section of the revised manuscript (please refer to line 204 in the unmarked version). We appreciate your insightful feedback, which has helped us improve the clarity and completeness of our paper.

5. Corrected for Multiple Comparisons:

Thank you for raising this important point. In our analysis (Figure 3b), where we compared major AKI factors across clusters, we focused on testing whether any of the labels significantly dominated within each cluster, rather than performing specific label-to-label comparisons. Hence, we did not adjust for multiple comparisons. We observed that clusters 3 and 4 demonstrated significant differences in label distributions, while clusters 1 and 2 showed distinct trends, though these were not statistically significant. We recognize the potential for misinterpretation of our initial statement regarding significant differences. Therefore, we have revised the applicable text to more accurately reflect our findings (please refer to line 211 in the unmarked version): " While there was a clear trend in the label distribution within each cluster, only clusters 3 and 4 showed statistically significant differences. The most dominant labels in each cluster were as follows: cluster 1, “Hypovolemia”; cluster 2, “Drug-related”; cluster 3, “Drug-related”; and cluster 4, “Cancer Cachexia." This approach was similarly applied in the survival analysis (Figure 3d) and the comparison of clinical characteristics (Table 2). The absence of multiple comparison adjustments in our analysis was due to our focus on demonstrating that SHAP-based clustering could reveal clinically relevant trends, rather than pinpointing specific differences between clusters. However, we acknowledge the importance of clearly communicating to our readers the lack of multiple comparison adjustments and appreciate your feedback regarding this matter.

Additionally, we would like to correct an error in our manuscript regarding the statistical test used for analyzing label distribution within each cluster. We incorrectly mentioned 'Cochran's Q test' in the Methods section; however, the actual test used was the 'Chi-square goodness-of-fit test'. We apologize for this oversight and have made the necessary correction in our manuscript.

6. Citation of Libraries and Code Language:

We appreciate your observation regarding the citation of the libraries and the programming language (Python) used in our study. Upon review, we realized that while we had included some citations, there were indeed areas where this information was insufficiently detailed. To address this, we have carefully revised the manuscript to ensure that all the libraries used are now properly cited and provided a more comprehensive acknowledgment of Python as the programming language. We believe these revisions more accurately reflect the resources utilized in our research and enhance the manuscript's clarity and transparency.

7. Addressing Missing Predictors and Independent Variable Impact:

Thank you for your insightful comment. In our research, we employed a combination of 11 unique features, 4 vital signs, and 68 types of time-series data, which included 52 laboratory values and 16 medication details, all extracted from Electronic Medical Records (EMRs). Detailed information on this can be found in our Supporting Methods (S2 Method) and Supporting Figure (S1 Fig.). We aggregated these features into lagged variables based on data from the preceding four weeks relative to each prediction point. Our focus was mainly on structured data with less than 50% missingness, but we acknowledge that unstructured data, such as notes from medical records and laboratory results with substantial missingness, could represent potential missing predictors. Furthermore, we ensured that each time-series data point was independently aggregated for each specific prediction time point. Consequently, each feature was used exclusively for the prediction relevant to its respective time point, maintaining the temporal independence of the variables. We appreciate your attention to this critical aspect of our methodology.

8. Inclusion of Patients with Multiple AKI Events:

Thank you for your insightful comment. Indeed, the period immediately following an AKI episode is generally recognized as a period carrying a significant risk for further AKI, and ordinarily, this period should not be excluded from study. However, in our study's model design, including the period right after the first AKI episode as part of the prediction target posed several challenges. Our model uses lagged variables, aggregating the average values for each week leading up to each prediction point. For example, the 'average Serum Creatinine (SCr) value of the last week' could vary significantly between the time just before and just after the initial AKI episode. This variation occurs because the rise in SCr value resulting from an AKI episode is reflected in the lag variables. In this case, the model would learn to label both the time point just before AKI and a time point after AKI as positive. Consequently, the model might learn the post-AKI increase in SCr value as an important predictive factor for AKI. This is particularly problematic if the AKI persists for several days, as the SCr value increase resulting from the first AKI episode would then significantly influence the lag variables used for learning AKI positive labels in subsequent days. As a result, the model could erroneously learn to prioritize the rise in SCr value as the main predictive factor for AKI. The aim of the AKI prediction model was to forecast future AKI before an increase in SCr values. Therefore, using the post-AKI rise in SCr for learning could misalign the model from its intended function. To maintain the accuracy of the model and prevent a decline in interpretability, we chose to exclude the period immediately following an AKI episode from the prediction target. Given that AKI can last several days, and considering that the aggregation is based on the average of the past week, we excluded the first two weeks after the initial AKI episode from our analysis.

Nevertheless, as you have rightly pointed out, the period immediately following the first AKI episode contains critical information about ongoing or worsening AKI. In our future research, we intend to analyze such conditions in more detail and develop models to predict persistent or worsening AKI, thereby addressing the areas not covered in this study. We are grateful for your valuable feedback.

9. Improvement of Figures and Heatmaps:

Finally, we acknowledge that the clarity of the figures and heatmaps was suboptimal. In response, we have enhanced the resolution and contrast of all figures, including the lines on heatmaps (2d and 2e), for better readability and interpretation.

Once again, thank you for your thorough review and insightful comments. We have endeavored to address each comment to the best of our ability and believe that as a result, our manuscript is now much improved. We are eager to hear any additional feedback that you may have.

Sincerely,

Yasushi Okuno, Ph.D. 

Department of Biomedical Data Intelligence, Graduate School of Medicine, Kyoto University 

53 Shogoin-Kawahara-cho, Sakyo-ku, Kyoto 606-8507, Japan 

okuno.yasushi.4c@kyoto-u.ac.jp

Motoko Yanagita, M.D., Ph.D.

Department of Nephrology, Graduate School of Medicine, Kyoto University

54 Shogoin-Kawahara-cho, Sakyo-ku, Kyoto 606-8507, Japan

motoy@kuhp.kyoto-u.ac.jp

 

Dear Reviewer #2,

Thank you very much for providing such detailed and constructive feedback on our manuscript. We have carefully considered each of your comment and have made corresponding revisions to enhance the quality and clarity of our study. We have taken your feedback seriously and have addressed each comment as follows:

1. Regarding the sentence in the introduction section, we have carefully reviewed our manuscript and were unable to find the specific sentence mentioned ("In recent years, AI has become an integral part of various industry verticals, and it is estimated that the market size of AI will reach 7.38 billion by 2025"). We would be grateful if you could kindly recheck this.

2. Similarly, in relation to the sentence about AI in healthcare in the discussion section, we also could not find this in our submission. We kindly request your assistance in verifying this.

3. We appreciate your constructive comments about the Methods section. We have expanded this section to provide more details on our data collection process, including the sources of data, types of data collected, and a comprehensive description of our data analysis methods. These enhancements, we believe, will significantly improve the understanding and credibility of our study.

4. Regarding the grammatical errors and typos, we are grateful for your attention to detail. We have thoroughly revised and proofread the manuscript to correct these issues, ensuring greater clarity and professionalism in our writing.

We are thankful for the opportunity to refine our manuscript based on your valuable feedback and hope that our revisions meet your expectations. Your guidance is crucial for the improvement of our work.

Sincerely,

Yasushi Okuno, Ph.D. 

Department of Biomedical Data Intelligence, Graduate School of Medicine, Kyoto University 

53 Shogoin-Kawahara-cho, Sakyo-ku, Kyoto 606-8507, Japan 

okuno.yasushi.4c@kyoto-u.ac.jp

Motoko Yanagita, M.D., Ph.D.

Department of Nephrology, Graduate School of Medicine, Kyoto University

54 Shogoin-Kawahara-cho, Sakyo-ku, Kyoto 606-8507, Japan

motoy@kuhp.kyoto-u.ac.jp

---

## [Decision Letter · Decision Letter 1]

30 Jan 2024

Interpretable machine learning-based individual analysis of acute kidney injury in immune checkpoint inhibitor therapy

PONE-D-23-28945R1

Dear Dr. Okuno,

We’re pleased to inform you that your manuscript has been judged scientifically suitable for publication and will be formally accepted for publication once it meets all outstanding technical requirements.

Kind regards,

Giuseppe Remuzzi

Academic Editor

PLOS ONE

Additional Editor Comments (optional):

Reviewers' comments:

Reviewer's Responses to Questions

**Comments to the Author**

1. If the authors have adequately addressed your comments raised in a previous round of review and you feel that this manuscript is now acceptable for publication, you may indicate that here to bypass the “Comments to the Author” section, enter your conflict of interest statement in the “Confidential to Editor” section, and submit your "Accept" recommendation.

Reviewer #1: All comments have been addressed

Reviewer #3: All comments have been addressed

2. Is the manuscript technically sound, and do the data support the conclusions?

Reviewer #1: Yes

Reviewer #3: Yes

3. Has the statistical analysis been performed appropriately and rigorously? 

Reviewer #1: Yes

Reviewer #3: Yes

4. Have the authors made all data underlying the findings in their manuscript fully available?

Reviewer #1: Yes

Reviewer #3: Yes

5. Is the manuscript presented in an intelligible fashion and written in standard English?

Reviewer #1: Yes

Reviewer #3: Yes

6. Review Comments to the Author

Reviewer #1: (No Response)

Reviewer #3: All the comments made by the first reviewers were addressed. I do not have any further comments. Well done.

7. PLOS authors have the option to publish the peer review history of their article (what does this mean?). If published, this will include your full peer review and any attached files.

Reviewer #1: **Yes: **Milou-Daniel Drici

Reviewer #3: **Yes: **Marcelo Rodrigues Bacci

---

## [Editor Report · Acceptance letter]

8 Mar 2024

PONE-D-23-28945R1 

PLOS ONE

Dear Dr. Okuno, 

I'm pleased to inform you that your manuscript has been deemed suitable for publication in PLOS ONE. Congratulations! Your manuscript is now being handed over to our production team.

Kind regards, 

on behalf of

Prof. Giuseppe Remuzzi 

Academic Editor

PLOS ONE